# Plasma-Treated Water: A Comparison with Analog Mixtures of Traceable Ingredients

**DOI:** 10.3390/microorganisms11040932

**Published:** 2023-04-03

**Authors:** Thomas Weihe, Yijiao Yao, Nevin Opitz, Robert Wagner, Johanna Krall, Uta Schnabel, Harald Below, Jörg Ehlbeck

**Affiliations:** 1Department of Plasma Biotechnology, Leibniz Institute for Plasma Science and Technology, 17489 Greifswald, Germany; 2Department of Food & Nutritional Sciences, University of Reading, Whiteknights, Reading RG6 6AD, UK; 3Institute for Hygiene and Environmental Medicine, Greifswald University Hospital, 17489 Greifswald, Germany; 4Center of Microbiology and Environmental System Science, Division of Terrestrial Ecosystem Research, University of Vienna, 1010 Vienna, Austria; 5Independent Researcher, 17489 Greifswald, Germany

**Keywords:** food safety, non-thermal plasma, atmospheric plasmas, plasma-activated water, reactive oxygen and nitrogen species, contaminations

## Abstract

Plasma-treated water (PTW) possess anti-microbial potential against *Pseudomonas fluorescence*, which is observable for both suspended cells and cells organized in biofilms. Against that background, the chemical composition of PTW tends to focus. Various analytical techniques have been applied for analyses, which reveal various traceable reactive oxygen and nitrogen compounds (RONS). Based on these findings, it is our aim to generate a PTW analog (anPTW), which has been compared in its anti-microbial efficiency with freshly generated PTW. Additionally, a solution of every traceable compound of PTW has been mixed according to their PTW concentration. As references, we treated suspended cells and mature biofilms of *P. fluorescence* with PTW that originates from a microwave-driven plasma source. The anti-microbial efficiency of all solutions has been tested based on a combination of a proliferation, an XTT, and a live–dead assay. The outcomes of the test proved an anti-microbial power of PTW that suggests more active ingredients than the traceable compounds HNO_3_, HNO_2_, and H_2_O_2_ or the combined mixture of the analog.

## 1. Introduction

Non-thermal plasmas (NTP) intensified appear as a decontamination technique and tend into the focus of research [1,2]. The energy of NTPs mainly houses in the free electrons, which results in a partial ionization of the gas and low gas temperatures [3,4]. However, a low gas temperature opens the possibility for plasma techniques to gain a foothold in new areas. NTP treatments can be roughly divided into direct (where a plasma source is directly acting on the target) and indirect treatments (where a plasma-treated media is used to treat the target) [5]. Plasma-treated water (PTW) is such an indirect method. It houses an anti-microbial activity, which is mediated by chemical reactions that were triggered during plasma-processed air (PPA) generation.

PTW houses a cocktail of active chemical compounds, which have been generated during its production. However, plasma-treated water (PTW) is such an indirect method. For the generation of PTW, different types of plasma sources are used, e.g., DBD, and plasma torches and jets. In addition, there is a huge variety of process schemes for the interaction of plasma with water [6]. In some applications, the water surface acts as an electrode; in others, the plasma has no direct contact with the water, and only the working gas, modified by the plasma interaction, is brought into contact with the water. In the present work is focused on PTW with high amounts of RNS. Therefore, microwave plasma sources were developed because of their ability to generate plasma in the temperature regime of 3.500 K, which is known to be very efficient in NOx generation. Due to their relatively high temperatures, microwave-induced plasmas push the reactions that are initiated in PTW into the direction of reactive nitrogen species. Therefore, PPA generation is often equated with the Birkeland–Eyde process [7]. However, since the Haber–Bosch process was invented, the Birkeland–Eyde process never played a dominant role. However, in the scope of a novel decontamination technique, nitrous gasses appear in a new light.

Whereas long-living compounds such as HNO_3_ or H_2_O_2_ are easy to analyze, many authors postulate short-living components of PTW and address them as active anti-microbial components. Additionally, several studies suggest a higher impact on the pH value of the solution than its ingredients. However, the composition of the PTW is not only governed by way of its production; the specific geometry of the device, the production temperature, or the type of plasma generation also influences the cocktail. PTW was able to demonstrate its anti-microbial effect on suspension cells and biofilms. Several authors describe the positive impact of PTW on foodstuffs [8,9,10,11]. For instance, Schnabel et al. (2021) describe the usage of PTW in a washing line of leafy greens [12]. However, more than just produce is anti-microbially processed with PTW. PTW is feasible for the cleaning of surfaces with a high anti-microbial decontamination rate [13,14]. It can be used as a bath for processing tools or as a sanitizer for wider, unspecific purposes [15]. Additionally, plasma appears in agriculture to increase agricultural production. For instance, Brust et al. (2020) used a dielectric barrier discharge system for the short-term treatment of cereal seeds [16]. The authors describe improved germination of the seeds. Plasma medicine is also an emerging field of plasma technology [17]. The anti-microbial potential opens new ways to combat pathogens such as bacteria or viruses. For instance, Guo et al. (2018) use PTW for efficient antiviral inactivation [18]. PTW also combats bacteria that gain antibiotic resistance and has the feasibility to open new perspectives and treatment schemes [19].

A growing number of researchers characterize PTW based on concentration measurements, pH [20], electron paramagnetic resonance (EPR) [21], or oxidative reaction potential [22]. At the same time, all the above-described methods gather information that describes the solution without being able to deduce a procedure for the mixing of PTW. Against that background, ionic chromatography and chronoamperometry appeared as ideal methods for the quantitative characterization of PTW. It is the aim of this study to generate a PTW analog (anPTW) from traceable compounds found in freshly produced PTW. The composition of the solutions is based on quantitative PTW analysis. The solutions, which are the anPTW, an HNO_3_-, an HNO_2_-, and an H_2_O_2_-solution, should be as potent in their anti-microbial activity as PTW. The decisions are based on a combination of three biological assays. This study presents an evaluation method where it compares every solution with PTW. The outcomes of the tests can be subdivided in six categories, which allows a classification of the solutions.

## 2. Materials and Methods

### 2.1. Experimental Design

It is the focus of the study to mirror the anti-microbiological impact of various solutions of nitrogen compounds (HNO_3_, HNO_2_) or hydrogen peroxide (H_2_O_2_) and a combined mixture of all compounds. The possible adverse impact of these solutions on *P. fluorescence* suspended cells, and *P. fluorescence* biofilms have been compared with the anti-microbial impact of freshly produced PTW. The impact of PTW on various microorganisms is sufficiently described in the literature [23,24,25]. The anti-microbial activity has been determined based on microbiological assays (colony forming unit (CFU), metabolic activity (XTT), and cell envelope integrity (live/dead)). Therefore, each mixed solution of HNO_3_, HNO_2_, H_2_O_2_, and analog PTW (anPTW, in the further course referred to as samples, x_data_) have been compared with the anti-microbiological impact of freshly produced PTW (in the further course referred to as reference, x_ref_). Each data point, which is a treatment of a cell suspension or a biofilm with PTW (x_ref_) or its analogs (anPTW, HNO_3_, HNO_2_, or H_2_O_2_), has been carried out with experimental 15 repetitions (N = 15).

### 2.2. PTW Production

PTW has been produced via a microwave-driven (2.45 GHz) plasma source. The MidiPLlexc is a torch-like source, which optimally works with a volumetric flow rate of 5 standard liters per minute (slm) and a treatment time of 900 s [26]. For the PTW generation, the plasma source has been mounted on a DURAN^®^ laboratory bottle (Schott, Mainz, Germany), and the efflux direction points towards the bottle base. Preliminary experiments showed an optimum volume of deionized water of 10 mL for a maximum of anti-microbial power after plasma-processed air acted on the water surface. As long as the volume did not exceed 10 mL, stirring was unnecessary and did not positively influence the ratio of surface/volume for an optimized gas uptake of the liquid.

### 2.3. PTW Characterization

#### 2.3.1. PPA Characterization by Fourier Transformation Infrared Spectroscopy (FTIR)

The Fourier Transform Infrared (FTIR, Vertex V70; Bruker, Billerica, MA, USA) spectroscopy has been employed to characterize the chemical compositions in PPA with 1 cm^−1^ resolution and an absorption path length of 0.25 m. Key reactive species in PPA were identified downstream of the plasma effluent under different operation conditions. The measurements were carried out at 1.5 m from the outlet of the plasma effluent generated by the MidiPLexc. The plasma was generated at a power of 50 W with an airflow of 1.5 slm. NO_2_, NO, and N_2_O_5_ were detected as long-living species in PPA (Figure 2a,b). As NO_2_ takes up more than 70% of the quantity of the key reactive species, it is, therefore, marked as an indicator for deciding optimal process parameters

Key reactive species in PPA were identified downstream of the plasma effluent under different operation conditions. The measurements were carried out at 1.5 m from the outlet of the plasma effluent generated by the MidiPLexc. The plasma was generated at a power of 50 W with an airflow of 1.5 slm. NO_2_, NO, and N_2_O_5_ were detected as long-living species in PPA. As NO_2_ takes up more than 70% of the quantity of the key reactive species, it is, therefore, marked as an indicator for deciding optimal process parameters.

#### 2.3.2. Ion Chromatographic Measurements

Traceable compounds of PTW have been determined via ion chromatography Professional 850 (Metrohm, Filderstadt, Germany) with a Metrosep A Supp 5-150/4.0 (Metrohm, Filderstadt, Germany) separation column and a Metrosep A Supp 5 Guard/4.0 (Metrohm, Filderstadt, Germany) guard column. The columns were operated at a temperature of 20 °C. Subsequent to the PTW production, a 100 mM solution of NaOH has been introduced. The NaOH stops the reaction of NO_2_ to NO_3_, triggered during the PTW production, by building stable NaNO_2_ salts. This reaction stop was achieved by mixing PTW and 100 mM NaOH in a ratio of 1:10. The samples were stored at 4 °C in an autosampler 899 IC Sample Center (Metrohm, Filderstadt, Germany). As the eluent, 3.2 mmol/L Na_2_CO_3_ and 2 mmol/L NaHCO_3_ with 10% acetonitrile with a flow rate of 0.7 mL/min were used. The detection of the solved ions was carried out via conductivity measurements and UV scans at 220 nm and 305 nm. Identification and the calculation of detected concentrations are based on calibration curves of NO_2_ and NO_3_ in a range of 10 to 100 mg/L. All experiments were repeated three times (N = 3), and each repetition was measured thrice.

#### 2.3.3. Chronoamperometry

For the H_2_O_2_ measurements, a screen-printed electrode (Drop-Sens, Llanera, Spain) with a surface covered with Prussian blue/glassy carbon has been used. The functionality of the chip electrode has been checked with a cyclic voltammogram in the range of −0.4–1.1 V. A typical pattern of the redox reaction at the chip electrode proves a calibrated voltammogram. The measurement was integrated into a flow system that continuously pumps a PBS buffer (pH 4.6, adjusted with 3 M KCl) into the measuring circuit with a rotational speed of 16 U/min. Due to the disintegration of H_2_O_2_ in alkaline surroundings, the pH of the buffer solution has to be below pH 7 [27]. At the outset, PTW was produced with the plasma device as described earlier. Subsequently, 500 μL of the sample was injected into the flow system and was measured in real-time. A potentiostat (Autolab PGSTAT101, Metrohm, Filderstadt, Germany) has been used. Data acquisition and evaluation have been performed on the software Nova1.10 (Metrohm, Filderstadt, Germany).

### 2.4. Mixtures of PTW Analogs

The PTW analogs are mixed based on the findings from the IC measurements and the chronoamperometry. For experiments with solutions only containing one compound found in PTW, HNO_3_, HNO_2_, and H_2_O_2_ have been diluted according to their concentration in freshly produced PTW. For the comparison between PTW and a solution as close to PTW as possible, all traceable PTW compounds have been mixed according to their concentrations in freshly produced PTW. The solution is called analogPTW (anPTW).

### 2.5. Microbiology

#### 2.5.1. Cultivation of *Pseudomonas fluorescence* Suspensions and Biofilms and Their Treatments with Sanitizing Agents

For the growth of suspension cells, *P. fluorescence* has been used (ATCC 13525, DSM 50090). Additionally, 1 L of a brain heart infusion broth (BHI, Carl Roth, Karlsruhe, Germany) has been prepared and autoclaved. The pH of the solution has been adjusted to pH 6. Subsequently, a single colony of a stock culture (Columbia agar, Carl Roth, Karlsruhe, Germany) was used to inoculate 50 mL of BHI, which were incubated for 24 h at 30 °C. On the following day, the suspension was adjusted with 0.8% NaCl to an optical density (OD) of 0.2 at a wavelength of 600 nm. The resulting suspension was transferred into the experiments. For treatment with a sanitizing agent, 1 mL of the diluted suspension was pipetted into a 15 mL Falcon^®^. Afterward, the presented suspension was supplemented with 1 mL of a sanitizing solution PTW (ref.), anPTW, HNO_3_, HNO_2_, and H_2_O_2_ (sample). Treatment times vary between 1 min to 5 min (1 min, 3 min, and 5 min). The reactions were stopped with 3 mL of PBS after the required treatment times (1 min, 3 min, and 5 min).

For the growth of *P. fluorescence* biofilms (ATCC 13525, DSM 50090), a was removed from the stock and transferred into 50 mL of BHI broth. This suspension was adjusted to an OD of 0.2 by a wavelength of 600 nm. Subsequently, 300 µL of the working suspension was pipetted into a well of a 96-well plate. Afterward, the 96-well plate was incubated at 30 °C for 24 h. On the following day, the supernatant was gently removed with a pipette. A gentle washing step followed, where PBS flushed the biofilms two times. Repeatedly, the *P. fluorescence* biofilms in each well were overlayed with 300 µL of BHI and incubated at 30 °C for another day. Here, we repeatedly followed the washing steps described above.

For the treatment, the biofilms were overlayed with a sanitizing agent for the desired treatment time (1 min, 3 min, and 5 min). Following the treatment, the sanitizing agents were removed, and the reactions were interrupted with 300 µL PBS. Subsequently, the biofilms were resuspended with a pipette tip, and the suspension was collected in an Eppendorf tube. The last step has been repeated, resulting in an overall sample volume of 600 µL. The suspensions have been used for CFU, XTT, and L/D assays.

#### 2.5.2. Proliferation Assay

A proliferation assay is a viable method that mirrors the proliferation ability of cells after PTW treatment or a treatment with its analogs (anPTW, HNO_3_, HNO_2_, H_2_O_2_). Against that background, 100 µL of the specimen was used for a serial dilution with dilution steps by a factor of 10 with maximum recovery diluent (MRD). The reference biofilms and treatment solutions have been diluted 1:10^12^. Additionally, 10 µL of each dilution have been pipetted onto a BHI plate and spread with the Miles and Misra technique with an adaption by tilting the plated spots. Subsequently, the plates were incubated at 30 °C for 24 h. Conclusively, the colonies were manually counted, and the *CFU*/mL values were calculated. Each countable colony has been interpreted as a *CFU*. Reduction factors (*RF*) have been calculated as follows:$$RF=log_{10}\left(CFU_{ref}\right)-log_{10}\left(CFU_{sam}\right)$$
with *RF* as the reduction factors, *CFU_ref_* as the *CFU*/mL of the reference, and *CFU_sam_* as the *CFU*/mL of the sample.

#### 2.5.3. XTT Assay

A colorimetric XTT assay (R&D-Systems, Minneapolis, MN, USA) determines the cell viability after a PTW treatment or a treatment with its analogs (anPTW, HNO_3_, HNO_2_, H_2_O_2_). Sample material was either adherent (biofilms) or suspended cells. The reagent kit reveals the cell viability as a function of the redox potential, which arises from a trans-membrane electron transport [28]. To trigger the intended reaction, N-methyl-dibenzopyrazine-methylsulfate (PMS) was used as an intermediate electron carrier. For the experiments, the 2,3-Bis(2-methoxy-4-nitro-5-sulfophenyl)-2H-tetrazolium-5-carbox-anilide (XTT) solution was diluted in a ratio of 1:50. Subsequently, the diluted solution undergoes a further dilution step when it has been mixed in a 96-well 1:3 with the sample. The prepared 96-well plate was incubated at 37 °C on a shaking table (80 rpm) for a maximal 24 h in a dark room. The incubated plates have been scanned at a wavelength of 470 nm with a plate reader (VarioskanFlash, Thermo Scientific, Waltham, MA, USA). The outcomes have been blank-corrected with the XTT-kit solutions at a wavelength of 670 nm. The outcomes of the experiments are specified relative to their references. In metabolically active cells, cleavage of the tetrazolium salt to formazan occurs via the succinate–tetrazolium reductase system based on bacterial dehydrogenases. The reaction is attributed mainly to bacterial enzymes and electron carriers [29]. The data of the assay are given relative to the reference (x_data_/x_ref_). At the same time, the references are obtained by XTT measurements of PTW-treated suspension cells or biofilms. This presents the following implications:(1)Data point < 1—a weaker anti-microbial effect than PTW (x_data_ < x_ref_);(2)Data point = 1—an anti-microbial effect as strong as PTW (x_data_ = x_ref_);(3)Data point > 1—a stronger anti-microbial effect than PTW (x_data_ > x_ref_).

#### 2.5.4. Live Dead Assay

The LIVE/DEAD BacLight^TM^ Bacterial Viability Kit (L7007, Invitrogen Detection Technologies, Carlsbad, CA, USA) is a two-color fluorescence assay of biological viability for a diverse array of bacterial genera. The kit uses a mixture of SYTO^®^ 9 green-fluorescent nucleic acid stain and the red-fluorescent nucleic acid stain, propidium iodide. These stains differ both in their spectral characteristics and in their ability to penetrate healthy bacterial cells. Used alone, SYTO^®^ 9 labels all cells, whether they possess an intact cell envelope or not. In contrast, propidium iodide solely penetrates bacteria with damaged envelopes, causing a reduction in the SYTO^®^ 9 stain fluorescence when both dyes are present. Thus, with an appropriate mixture of SYTO^®^ 9 and propidium iodide stains, bacteria with intact envelopes stain with a fluorescent green, whereas bacteria with damaged envelopes stain with a fluorescent red [30,31].

In a practical application, 0.9µL of the mixture have been added to 300 µL of every sample solution into a 96-well plate. In a dark room at 25 °C, the solutions were incubated on a rotary shaker for 20 min. The fluorescence of each 96-well plate was determined with a fluorescence microplate reader (VarioskanFlash, Thermo Scientific, Waltham, MA, USA) at an excitation wavelength of 485 nm and emission wavelength of 530 nm (green, G) and 630 nm (red, R). Subsequently, a G/R ratio was calculated and used for data evaluation. All L/D assay results are specified relative to their reference. Values < 1 reflect a decreased outcome relative to the reference, and a value > 1 reflects an increased outcome relative to the reference. The data of the assay is given relative to the reference. At the same time, the references are obtained by L/D measurements of PTW-treated suspension cells or biofilms. This presents the following implications:(1)Data point < 1—a weaker anti-microbial effect than PTW (xdata < xref);(2)Data point = 1—an anti-microbial effect as strong as PTW (xdata = xref);(3)Data point > 1—a stronger anti-microbial effect than PTW (xdata > xref).

### 2.6. Statistics

#### 2.6.1. Statistical Tests

All data shown in the graphs represent the mean and the standard deviation. The mean and the standard deviation include a sample size of N = 15. All values obtained from the proliferation assay are presented as the decadic logarithm (log_10_) of the CFU counts. The normality of the logarithmic values and the results of the XTT- or L/D assays have been tested through a Shapiro–Wilk test (*p* > 0.01). Additionally, the results were visualized by QQ plots. Decisions regarding successful anti-microbial effective treatment were based on multivariate analysis of variance (MANOVA). The results of the MANOVA mirror the difference between two compared treatments with two different solutions (i.e., PTW vs. HNO_3_). An analysis of variance (ANOVA) tries to explain the differences between various treatment times. The significance level has been set to α = 0.05. Statistically different treatment groups (1 min, 3 min, and 5 min) have been graphically determined based on a post-hoc procedure.

#### 2.6.2. Data Evaluation

For both suspension cells and biofilms, successful PTW treatments have been evaluated through a combination of three assays. A proliferation assay, based on a plate count method, mirrors the count of viable *P. fluorescence* either suspended or hosted in a biofilm. It is the kind of cell that has an active metabolism in a suspension or contributes to the active anabolism of a biofilm. A drop in the number of those cells represents a decreased number of cells that are able to reproduce themselves and live under ideal growth conditions. The colorimetric XXT assay mirrors the metabolic activity of the viable cells. According to the rule, a decreased CFU is followed by a decreased XTT signal. The third test in the row is a live–dead assay (L/D), which mirrors the vitality of the cells based on the integrity of the cell membrane. If cells are killed in a PTW decontamination step, proliferation (CFU), metabolic activity (XTT), and membrane integrity (L/D) are decreased in a comparable magnitude (cf. Figure 1a). If there is no drop in the L/D assay in addition to decreased values for CFU and XTT, cells enter the viable but nonculturable state (VBNC).

Figure 1a summarizes the possible and microbiologically meaningful outcomes for the assay combination of a CFU, XTT, and an L/D assay. As shortly mentioned above, a decrease in all three outcome values mirrors the killing of suspension cells and the killing of parts or the detachment of viable cells of the biofilm. Cells were pushed into the VBNC state when CFU and XTT decreased their outcome values, and no statistically meaningful changes in the L/D assay appeared. A detachment of nonviable cells appears when the cells show a lower proliferation in the CFU assay and a value decrease for XTT and L/D. If an increased proliferation is flanked by ambiguous outcomes with a tendency for increased values, it will rather be interpreted as cross-contamination. Microbiologically non-meaningful outcomes reflect a failed experiment.

The decision of whether an anti-microbial treatment with one of the solutions was successful or not is judged based on the outcome combination of three test kits. Arrows in a tabular scheme mirror an increase, a decrease, or an invariable outcome in either one of the tests. The decision of whether an outcome is interpreted as an increased, decreased, or invariable value is judged based on statistical tests (MANOVA). Figure 1b summarizes the statistic classification criteria for the arrow directions. If the compared solutions in Figures 3 and 4 both show the same tendency but in a different magnitude, the values of both solutions have been compared via ANOVA. A rejected null hypothesis is interpreted as a definitive anti-microbial action of the solution that starts at t = 1 min and ends up at another time point. Down and up pointing arrows show a rejected null hypothesis based on a graphical evaluation of the MANOVA outcomes. If the null hypothesis cannot be rejected, the outcome will be interpreted as a failed process without any changes in the cell density in the suspensions or the biofilms.

## 3. Results

### 3.1. PTW and Its Compounds

The generation of plasma-processed water (PTW) started with compressed air carried over a microwave-driven plasma torch. Subsequently, the air was processed in a manner dependent on the used plasma source. When microwave sources were used, predominantly reactive nitrogen species and, to a lesser extent, reactive oxygen species were produced. Subsequently, the PPA has carried over pipes, where it cools down. Consequently, PPA processed via a microwave source houses nitroxides pushed into a metastable state. Traceable, long-living compounds of PPA were determined via Fourier Transform Infrared spectroscopy (FTIR). Figure 2a summarizes the findings of the measurements. NO_2_ was detected in the range of 1540–1660 cm^−1^, with the second order of energy absorption in the range of 2840–2938 cm^−1^. NO was observed in the range of 1790–1950 cm^−1^, while N_2_O_5_ was detected in the 700–779 cm^−1^, 1220–1292 cm^−1^, and 1668–1791 cm^−1^ ranges. Due to the existence of water under high humidity operation conditions, a small amount of nitric acid was produced and observed in the spectra of PPA. As follows, NO_2_ takes up more than 70% of the quantity of the key reactive species; it is, therefore, marked as an indicator for deciding optimal process parameters.

PTW is a chemical cocktail of compounds that originates from chemical reactions started by the action of a plasma source. As is observed for PPA, the compounds found in the processed water originate from the metastable and long-lived RONS. Those traceable and long-living species were determined via ion chromatography and potentiometric measurements. Table 1 summarizes the findings of the analytical measurements of PTW.

Figure 2b summarizes the postulated and/or literature findings of possible reactions in PTW. The most prominent compound found in PTW is HNO_3_ (33mol/L). Additionally, we observe a concentration of 1.17 mol/L of HNO_2_. Both compounds can be generated from NO_2_ of PPA. The reactive oxygen species H_2_O_2_ was also found in freshly generated PTW (21.1 mol/L). All compounds monitored in PTW have been solved in water.

### 3.2. Anti-Microbial Activity of PTW against Suspension Cells of Pseudomonas fluorescence

Figure 3 shows the outcomes for the treatment of suspension cells with PTW, anPTW, HNO_3_, HNO_2_, and H_2_O_2_. Figure 3 upper part shows all data of all solutions merged in one sketch. The graph represents the magnitude of all values relative to each other. Figure 3 lower part reflects the comparison of the anti-microbial effect of PTW with the effect of solutions of its ingredients. Additionally, the impact of PTW on suspension cells is compared with the impact of its analog (anPTW) made from the traceable compounds found in IC- and potentiometric measurements.

For suspended *P. fluorescence* cells in a proliferation assay (CFU), the action of PTW after 1 min treatment time compared with the impact of anPTW after the same treatment time show no statistically meaningful difference for their RFs (PTW vs. anPTW, RF, 1 min: 2.61 ± 0.52 vs. 2.14 ± 1.31, ANOVA: *p* < 0.05). Nevertheless, both treatments reveal a statistically meaningful decrease in the cell numbers in that assay (ANOVA: *p* < 0.05). The observation continues for a 3 min and 5 min treatment, where no further meaningful changes (ANOVA: *p* > 0.05) appear for both kinds of treatments (PTW vs. anPTW, RF, 3 min: 2.91 ± 0.62 vs. 3.14 ±1.11; 5 min: 3.04 ± 0.75 vs. 3.67 ± 0.70). The XTT assay follows that tendency, and after a drop (PTW and anPTW, MANOVA: *p* < 0.05) of the metabolic activity of the cells for both treatment solutions, no further statistically meaningful changes (PTW and anPTW, ANOVA: *p* > 0.05) have been determined (PTW vs. anPTW, 3 min: 0.32 ± 0.86, vs. 0.19 ± 1.39, 5 min: 0.45 ± 0.21 vs. 0.22 ± 0.09). The L/D assay for PTW underpins the interpretation of the CFU- and XTT assay. A statistically meaningful drop (PTW, ANOVA: *p* < 0.05) of the membrane integrity is observed after treatments of 1 min, which show no further meaningful (PTW, ANOVA, *p* < 0.05) drops for longer treatments. anPTW treatments show no statistically meaningful impact on the membrane integrity (anPTW, ANOVA, *p* < 0.05). A treatment with anPTW pushes the cells into VBNC, whereas a PTW treatment appears successful.

Treatment of suspended cells with HNO_3_ appears ineffective when compared with that of a PTW treatment. First, no statistically meaningful (HNO_3_, ANOVA, *p* < 0.05) changes in the cell count are observed after a 1 min treatment (HNO_3_, RF, 1 min: −0.04 ± 0.06); a tendency that continues after a 3 min and 5 min treatment time (HNO_3_, RF, 3 min: 0.06 ± 0.07, 5 min: −0.04 ± 0.08). There are no statistically meaningful changes in cell number (HNO_3_, ANOVA: *p* > 0.05). The L/D assay completes the picture and now shows statistically meaningful changes (HNO_3_, ANOVA: *p* > 0.05) in the relation between living and dead cells (HNO_3_, all values are relative to their reference, 1 min: 1.33 ± 0.74, 3 min: 1.27 ± 0.81, 5 min: 1.29 ±0.79). All PTW-induced changes in the cell number, their metabolic activity, and their membrane activity appear statistically meaningful when compared to the ones obtained after an HNO_3_ treatment. An HNO_3_ treatment was judged as an ineffective failure.

An HNO_2_ treatment of suspended cells appears effective but to a lesser extent. At the same time, the CFU count is not statistically meaningful or different from that of a PTW treatment (PTW vs. HNO_2_, ANOVA: *p* > 0.05). The means of the cell numbers are quite different (PTW. vs. HNO_2_, RF, 1 min: 2.61 ± 0.52 vs. 1.30 ± 1.11), but huge error bars suffer the mean of the CFU number. Both treatments show no further statistically meaningful (PTW and HNO_2_, ANOVA, *p* < 0.05) impact for longer treatment times (HNO_2_, RF, 3 min: 1.09 ± 0.95, 5 min: 0.66 ± 1.41). The XTT assay follows that tendency, and no meaningful changes (HNO_2_, ANOVA: *p* < 0.05) were determined throughout the whole experimental series (HNO_2_, all values are relative to its reference, 1 min: 0.51 ± 0.18, 3 min: 0.44 ± 0.21, 5 min: 0.45 ± 0.21). The L/D assay also follows the tendency, and again, there were no statistically meaningful changes (HNO_2_, ANOVA: *p* < 0.05) of the cell integrity due to an observed HNO_2_ treatment (HNO_2_, all values are relative to their reference, 1 min: 0.12 ± 0.83, 3 min: 0.19 ± 0.72, 5 min: 0.15 ± 0.68). An HNO_2_ treatment seems to detach dead cells, which leads to a drop in the cell number but to rather invariable values for the metabolic activity or the integrity of the cell membrane of *P. fluorescence*.

The H_2_O_2_ treatment of suspended cells has a CFU drop after a 1 min treatment of a comparable magnitude of a PTW treatment (PTW vs. H_2_O_2_, RF, 1 min: 2.61 ± 0.52 vs. 2.72 ± 0.56) and no statistically meaningful difference (PTW vs. H_2_O_2_, ANOVA, *p* < 0.05) could be detected. Usually, longer treatment times lead to a higher RF. Contrarily, an H_2_O_2_ treatment reveals decreased RFs for treatment times of 3 min and 5 min (H_2_O_2_, RF, 3 min: 0.63 ± 0.61, 5 min: 0.98 ± 1.03). The changes appear to be statistically meaningful (H_2_O_2_, ANOVA: *p* < 0.05). Unlike the proliferation assay, the XTT assay shows no meaningful (H_2_O_2_, ANOVA: *p* > 0.05) decreased metabolic rate due to an H_2_O_2_ treatment relative to their references. The L/D assay depicts no statistically meaningful (H_2_O_2_, ANOVA: *p* > 0.05) drop in the cell integrity for all treatments (H_2_O_2_, all values relative to their references, 1 min: 0.83 ± 0.22, 3 min: 0.72 ± 0.174, 5 min: 0.68 ± 0.19). The result is interpreted as the detachment of dead cells.

### 3.3. Anti-Microbial Activity of PTW against Biofilms of Pseudomonas fluorescence

Figure 4 shows the outcomes for the treatment of biofilms of *P. fluorescence* with PTW, anPTW, HNO_3_, HNO_2_, and H_2_O_2_. Figure 4 upper part shows all data of all solutions merged in one sketch. The graph represents the magnitude of all values relative to each other. Figure 4 lower part reflects the comparison of the anti-microbial effect against biofilms of PTW with the effect of solutions of its ingredients. Additionally, the impact of PTW is compared with the impact of its analog (anPTW) made from the traceable compounds found in IC- and potentiometric measurements.

For biofilms of *P. fluorescence* in a proliferation assay (CFU), the action of PTW after 1 min treatment time compared with the impact of anPTW after the same treatment time showing no statistically meaningful difference (MANOVA, *p* > 0.05) for their RFs (PTW vs. anPTW, 1 min: 3.97 ± 0.51 vs. 5.74 ± 2.79). The observation continues for a 3 min treatment. As opposed to this, a statically meaningful difference (*p* < 0.05) is observed for a 5 min treatment (PTW vs. anPTW, 5 min: 3.73 ± 0.62vs. 8.08 ± 1.77). The XTT assay draws a picture and vice versa. Both solutions show a drop in their metabolic activity when compared with their reference (PTW vs. anPTW, 1 min: 0.26 ± 0.33 vs. 0.62 ± 0.22). For both solutions, these changes are statistically meaningful (MANOVA, *p* < 0.05). Based on the drop (ANOVA, *p* < 0.05) of the metabolic activity after a 3 min PTW treatment (PTW, 3 min: 0.19 ± 0.18), the PTW treatment is interpreted to have a stronger adverse effect on suspended cells and biofilms than anPTW. Both solutions show no further statistically meaningful (MANOVA, *p* > 0.05) drop in the metabolic activity for longer treatments. The L/D assay for PTW underpins the interpretation of the CFU and XTT assay. A statistically meaningful drop of the membrane integrity is observed after a treatment of 1 min (PTW, 1 min: 0.29 ± 0.14), which shows no further meaningful (*p* < 0.05) drops for longer treatments. After a 1 min anPTW treatment of suspended *P. fluorescence* cells and biofilms, no statistically meaningful change of the cell integrity is observed (anPTW, 1 min: 0.84 ± 0.16). However, the drop of the membrane integrity became meaningful after an anPTW treatment of 5 min (anPTW, 5 min: 0.62 ± 0.11). A treatment with anPTW may be successful after 5 min, but at least a 1 min treatment pushes the cells into VBNC.

The comparison of the PTW treatment vs. HNO_3_ is unambiguous. The 1 min PTW treatment shows an RF of 3.97 ± 0.51 vs. an RF of 0.21 ± 0.37 after an HNO_3_ treatment. For both treatment solutions, no statistically meaningful changes have been observed for longer treatments (PTW vs. HNO_3_, 3 min: 4.05 ± 0.65 vs. 0.25 ± 0.29, 5 min: 3.73 ± 0.62 vs. 0.35 ± 0.34, ANOVA: *p* < 0.05). The trend continues for the XTT assay where a statistically meaningful drop of the metabolic activity is observed after a 1 min PTW treatment (PTW relative to its reference, 1 min: 0.26 ± 0.33, ANOVA: *p* < 0.05). No statistically meaningful further changes for longer treatments were revealed (PTW relative to its reference, 3 min: 0.19 ± 0.18, 5 min: 0.34 ± 0.21, ANOVA: *p* > 0.05, ANOVA: *p* > 0.05). An HNO_3_ treatment of biofilms shows no statistically meaningful (*p* > 0.05 for ANOVA) changes at any treatment times (HNO_3_ relative to its reference, 1 min: 0.91 ± 0.21, 3 min: 0.88 ± 0.23, 5 min: 1.01 ± 0.07). The L/D assay makes the same statement. While a meaningful (ANOVA: *p* < 0.05) drop in the cell integrity of *P. fluorescence* biofilms is revealed after a 1 min PTW treatment (PTW relative to its reference, 1 min: 0.29 ± 0.15), an HNO_3_ treatment showed no meaningful drop (HNO_3_ relative to its reference, 1 min: 0.86 ± 0.16, ANOVA: *p* > 0.05). The tendency is conserved for longer treatments (PTW vs. HNO_3_, all values relative to their references, 3 min: 0.26 ± 0.14 vs. 1.09 ± 0.29, 5 min: 0.25 ± 0.15 vs. 0.93 ± 0.27) and no statistically meaningful changes have been observed (PTW, ANOVA: *p* > 0.05; HNO_3_, ANOVA: *p* > 0.05). The successful PTW treatment opposes a failure in terms of the anti-microbial action of the HNO_3_ treatment.

The comparison of the anti-microbial active PTW treatment with those of an HNO_2_ treatment reveals statically meaningful RF for both treatments (PTW vs. HNO_2_, 1 min: 3.97 ± 0.51 vs. 5.65 ± 2.71) but no statistical difference (ANOVA: *p* > 0.05). Nevertheless, they become meaningful after a 3 min and 5 min treatment (PTW vs. HNO_2_, all values relative to their references, 3 min: 4.05 ± 0.65 vs. 8.88 ± 3.97, 5 min: 3.73 ± 0.62 vs. 9.15 ± 1.63). The XTT assay shows no meaningful changes for an HNO_2_ treatment for all treatment times (HNO_2_, ANOVA: *p* > 0.05). The PTW treatment has a meaningful (see above) drop in metabolic activity, which shows no further impact for longer treatment times. The comparison of a 5 min PTW treatment with an HNO_2_ treatment of the same length shows no meaningful difference in their outcomes (ANOVA: *p* < 0.05). However, since a PTW treatment shows a statically meaningful change for a treatment up to 5 min (ANOVA: *p* < 0.05), we interpret these findings as decreased cell integrity due to a PTW treatment and rather invariable values or only light changes for an HNO_2_ treatment (HNO_2_ relative to its reference, 3 min: 0.87 ± 0.31, 5 min: 0.69 ± 0.37). The HNO_2_ treatment with its high RFs in the proliferation assay is interpreted as a detachment of dead cells with a possible small number of living or VBNC cells of *P. fluorescence*.

An H_2_O_2_ treatment reveals, when compared with a PTW treatment of the same manner, meaningful changes (ANOVA: *p* < 0.05) in their cell count (H_2_O_2_, RF, 1 min: 5.12 ± 1.96) after a treatment time of 1 min. No further meaningful (H_2_O_2_, ANOVA: *p* > 0.05) changes are observed for longer treatment times (H_2_O_2_, RF, 3 min: 5.12 ± 2.97, 5 min: 4.13 ± 2.28). Both the XTT and the L/D assays show no statistically meaningful (ANOVA: *p* > 0.05) decrease for the metabolic activity of the cells (H_2_O_2_ relative to their references, XTT, 1 min: 0.74 ± 0.26, 3 min: 0.92 ± 0.21, 5 min: 1.01 ± 0.0) or an adverse impact of an H_2_O_2_ treatment on the cell integrity (H_2_O_2_ relative to their references, L/D, 1 min: 0.86 ± 0.21, 3 min: 0.91 ± 0.15, 5 min: 1.01 ± 0.29), respectively. The results have been interpreted as a VBNC state of an H_2_O_2_ treatment opposing the successful PTW treatment.

## 4. Discussion

PTW is produced by means of PPA. That is, ordinary compressed air is processed with an ignited plasma torch, which is subsequently passed into distilled water. This step initiates chemical reactions that are at the beginning of every PTW action. Generally, the choice of a certain plasma source brings the opportunity to push chemical reactions on the desired path. For non-thermal plasmas (NTP), the main conversion rates of certain chemical reactions are either on the side of reactive oxygen or on the side of reactive nitrogen compounds [24,32]. In the case of microwave-generated plasma, the main conversion rates for atmospheric compounds are on the oxidation of nitrogen [2]. A process that resembles a Birkeland–Eyde process, which is basically a plasma process using an electric arc [7]. The process has been used for the production of nitrous gases as a starting product for several products of the chemical industry, such as ammonia, fertilizer, or explosives [7,33].

Directly after the processing of the compressed air, the freshly produced PPA was studied by FTIR measurements, and the radical NO_2_ was the main component that was detectable. Additionally, the gaseous NO has been observed. In addition to that finding, the spectra reveal the presence of the solid N_2_O_5_. However, N_2_O_5_ has a sublimation point that is very close to room temperature (32 °C). Against that background, it is likely to detect sublimated N_2_O_5_ in the gaseous PPA. It is surprising that no N_2_O_4_, the dimeric of NO_2_, was observed in the spectra. Most likely, the concentration is too low. The IR-active chemical compounds have been the expected outcome for such a measurement [34]. The measurements underpin the statement of the similarity of the Birkeland–Eyde process and the presented PPA generation via a microwave-driven plasma source [7].

Subsequently, the PPA is passed into distilled water, and at the liquid–gas interface, gaseous reactive nitrogen species enter the liquid [24]. Consequently, microwave-generated PTW houses a set of nitrous compounds that provide anti-microbial efficacy [35]. The study aimed to answer whether it is possible to mix anPTW from the results of analytical measurements such as IC, pH measurements, or electrochemical characterizations of the PTW. As a first step, IC measurements revealed a high content of nitrous and nitric acid. Nitrous acid appeared as the main component of the PTW solution and resulted from the reaction of nitrogen dioxide, which entered the solution as a part of PPA, with water [4]. Additionally, NO_2,_ in a reaction with H_2_0, is also a source for HNO_3_. However, we observe different concentration ratios for HNO_2_ and HNO_3_, which can be greater or less than 1 depending on the plasma source used. Further PTW reactions, which are initiated by passing PPA, solely postulate the presence of peroxynitric acid (HNO_4_) or the conjugated base peroxynitrite (OONO^−^), which is a short-living component, and its detection appears challenging. In the literature, it is also described as a reservoir for NO_2_ through a reversible radical reaction [36]. However, the characterization of metastable (in addition to N_2_O) nitrous gases is not a new field of research, and many scientists gained knowledge through their research. Since nitrous compounds are ubiquitous, researchers from many fields enter the scene [36,37,38]. Additionally, cracked and oxidized nitrous compounds are pushed into a metastable state due to cooling down after the process. As early as the generation of PPA, the relative composition of PTW strongly depends on the composition of PPA, which again depends on the generation temperature and the subsequent cooling down influenced by the source set up [32]. Generally, the composition of PTW depends on many factors such as the type of the source, excitation properties, or the gas flow rate [39]. Against that background, it is advantageous to consider the total content of all nitrogen compounds and not individual components [40,41].

Although only H_2_O_2_ has been detected, PTW also houses reactive oxygen species [42]. Due to a higher power density of a microwave-induced plasma, the process is performed at higher temperatures as reached by radiofrequency plasmas. Against that background, it is rather likely to find a plasma gas containing a higher proportion of reactive nitrogen species than reactive oxygen species [9]; PTW generates the reactive components of PPA, which is predominantly NO and NO_2._ Theses compounds react to HNO_3_ and HNO_2_ when introduced into PTW. Beside those reactions, H_2_O_2_ have been also found in PTW: However, the H_2_O_2_ concentration was higher than the HNO_2_ concentration but much lower than the HNO_3_ concentration. It is an expected result for PTWs produced with a microwave-induced plasma device.

It was the aim of the study to answer the question of whether it is possible to mix an analog PTW out of commercially available chemicals. More importantly, are these solutions as anti-microbial efficient as freshly produced PTW? Against that background, a series of assays (CFU, XTT, and L/D) have been performed on suspension cells and biofilms of *P. fluorescence*. The assays monitor cell proliferation (CFU), metabolic cell activity (XTT), and cell integrity. Since all tests have been performed in a dynamic time frame, the results of the assays are suffered by high standard deviations. To answer the question posed at the beginning of the chapter, we statistically evaluated the results in terms of a direct comparison of PTW with every traceable ingredient found in freshly produced PTW. The anti-microbial success of treatment with one of the solutions is judged on an arrow system described earlier.

An anti-microbial treatment against suspended *P. fluorescence* and mono-species biofilms with PTW appears successful. We see a drop in the proliferation followed by a decreased metabolic activity and results from the L/D assay lower than one, which is, in combination with the two other tests, a drop in the number of living cells. When such treatment is compared with anPTW, the anPTW treatment appears as effective as a PTW treatment since we see a decreased proliferation. However, when all three assays are taken into account, no decreased cell integrity is observed. Against that background, it is likely to assume more cells have entered VBNC. A decreased proliferation and metabolic activity speak for the mortality of these cells. Contrarily, for killed suspension cells, an L/D value lower than one can be expected. A test–outcome combination very likely speaks for cells entering the VBNC state.

A comparison with HNO_3_, which is a concentrated compound found in PTW, did not show any anti-microbial efficiency for both suspension cells and biofilms. No meaningful changes are observed for the whole assay series. The result is interpreted as an unsuccessful treatment. The result is rather confusing since HNO_3_ is frequently used in CIP procedures to combat adverse biofilms [43]. However, since CIP procedures work with different solutions from the alkaline to the acidic end of the spectra, it is rather likely that a combination of active agents leads to success. In addition to that, although the concentration of HNO_3_ is relatively high in PTW, higher concentrations may have a more pronounced effect on microorganisms.

A comparison of HNO_2_ action on suspended cells and biofilms of *P. fluorescence* reveals a process that pushes suspended cells into VBNC and detaches dead cells from the biofilm. The arguments for interpreting the outcomes of the combined tests as VBNC do not differ from the ones given before. The detachment of cells follows from a decreased proliferation framed by no statistically meaningful changes in the metabolic rates and the integrity of the cells. Compared with a PTW treatment, an HNO_2_ treatment appears less effective. Especially if only the CFU is taken into account, the treatment is less effective against suspension cells but more effective when biofilms are combated. The XTT assay of HNO_2_-treated cells and biofilms varies for all treatment times and seems to be decreased, but huge error bars suffer it. Here, no statistically meaningful changes could be detected. Overall, an HNO_2_ treatment pushes suspended cells into VBNC and detaches biofilms. Nevertheless, VBNC can be interpreted as a cellular stress reaction induced by stress such as starvation, osmotic or oxidative stress, or adverse nutrition [44,45,46,47]. Stress is always a matter of an offensive descent on bacterial cells. Is it only one or a few components that mediate the stress, or is a cell very likely to be in the position to defend itself? Nevertheless, if, as is the case for PTW, the solutions have a potpourri of different anti-microbial components, it becomes likely that a cell lacks sufficient defense mechanisms, which is a stress that treatment with HNO_2_ could well provide. Nevertheless, it does not provide the anti-microbial efficiency of PTW, which rather kills the cells instead of pushing them into VBNC. For biofilms, a cell detachment could be as effective as the killing of the cells, but it houses the danger of cross-contamination.

Treatment of suspended cells of *P. fluorescence* with H_2_O_2_ equals an H_2_O_2_ treatment of biofilms of *P. fluorescence*. Both H_2_O_2_ treatments push suspended cells and biofilms into the VBNC state. For both manifestations of *P. fluorescence*, slight changes in the L/D assay are visible. Albeit they are not statistically meaningful, the changes depict a contrary effect. Whereas a decrease in cell integrity with longer treatment times is observed for suspension cells, it seems to increase with longer treatment times for biofilms. H_2_O_2_ treatments seem to be successful but show no meaningful results. The H_2_O_2_ mechanism of action is interpreted as cellular stress, which might be opposed by cellular protection mechanisms [48,49]. An H_2_O_2_ treatment has been interpreted as less effective than a PTW treatment.

If all outcomes of the treatments against suspended cells and biofilms of *P. fluorescence* are added up, PTW appears as the most powerful anti-microbial solution among the tested ones. Either a solution of a diluted component found in PTW or anPTW could fully explain the anti-microbial efficiency and its magnitude. It is obvious that PTW houses a complex chemistry, which is not explained by the determination of traceable long-living species. Several authors postulate various short-living species, such as peroxynitrite or even complex water structures in PTW [41,50,51]. Deeper future investigations need to explain the excellent anti-microbial potential of PTW. Additionally, Niquet et al. (2017) chemically compare DBD and microwave-generated PTW and the biological effect of the compounds found in these solutions [24]. These authors demonstrate how the concentrations of the compounds are dependent on the type of generation. Against that background, our study demonstrates for the first time that PTW has better anti-microbial activity than a mixture of its detectable components. The exact mechanism of PTW action needs to be clarified in future studies.

## 5. Conclusions

Is it possible to mix an anPTW that is as anti-microbially effective as PTW from the results of traceable compounds of a deeper PTW analysis? It was the aim of this study to answer this question. Based on our findings, anPTW shows RFs, which are comparable (suspension cells) with or exceed (biofilms) those of PTW. Nevertheless, with the combination of a CFU, an XTT, and an L/D assay taken into account, PTW appears as the only solution (when compared with anPTW, HNO_3_, HNO_2_, and H_2_O_2_) that successfully combated suspension cells and biofilms.

## Figures and Tables

**Figure 1 microorganisms-11-00932-f001:**
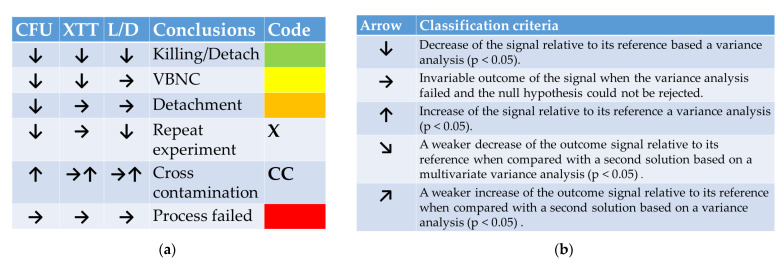
(**a**) Interpretation scheme for Figures 3 and 4 listed in a tabular form. Relative to its reference, a down-pointing arrow shows a decreased assay outcome, a horizontal arrow mirrors, and an up-pointing arrow stands for an increased test outcome. (**b**) Decision scheme for the classification criteria of a PTW treatment impact on cell viability.

**Figure 2 microorganisms-11-00932-f002:**
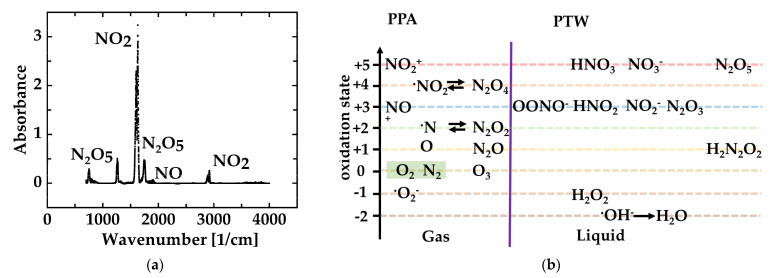
The figure summarizes the theoretical background, which delivers the assumptions made for the interpretation of the PTW mechanisms of actions: (**a**) Interpretation scheme for combined testing of what is contained in the first panel; (**b**) Overview of compounds and their oxidation steps found in PPA in FTIR measurements and in analytics of PTW.

**Figure 3 microorganisms-11-00932-f003:**
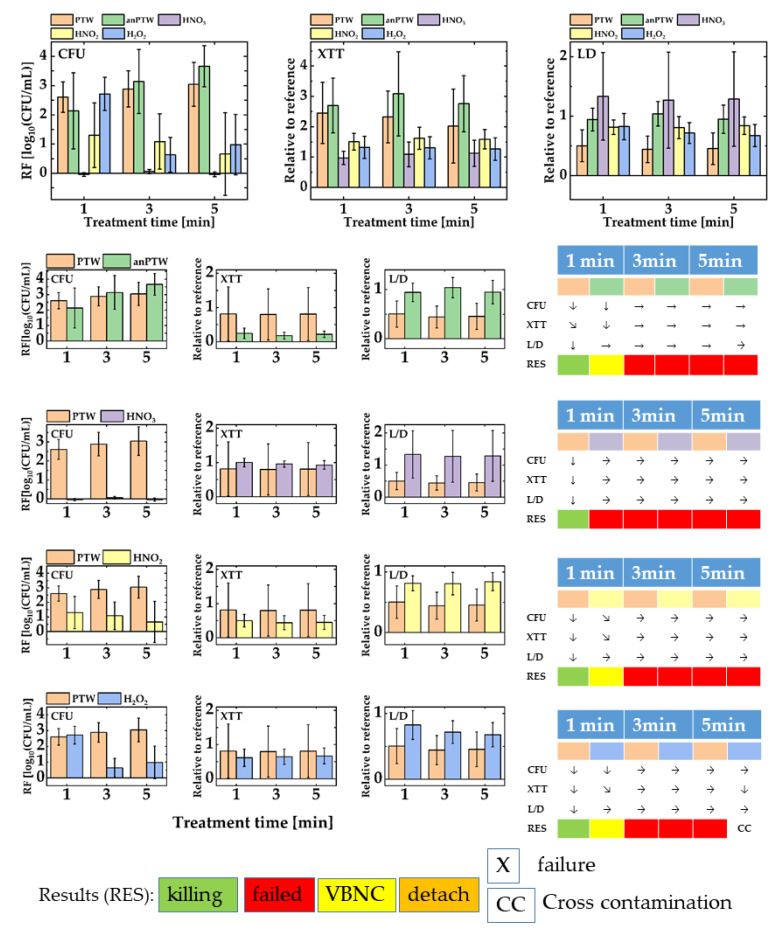
A summary of all assay data (CFU, XTT, and L/D) for suspension cells of *Pseudomonas fluorescence.* The upper part of the figure gathers all data from all sanitizing solutions (PTW, anPTW, HNO_3_, HNO_2_, and H_2_O_2_). The lower part of the sketch compares every sanitizing solution versus freshly produced PTW. The result of treatment with such solutions has been underpinned with an assay evaluation system as described earlier (Materials and Methods Section 2.6.2). For convenience, only statistically meaningful changes have been taken into account. In a series of independent ANOVA (*p* < 0.5), the results of a certain treatment time have been compared with those of the previous point of the same data set (1 min vs. reference, 3 min vs. 1 min, and so on).

**Figure 4 microorganisms-11-00932-f004:**
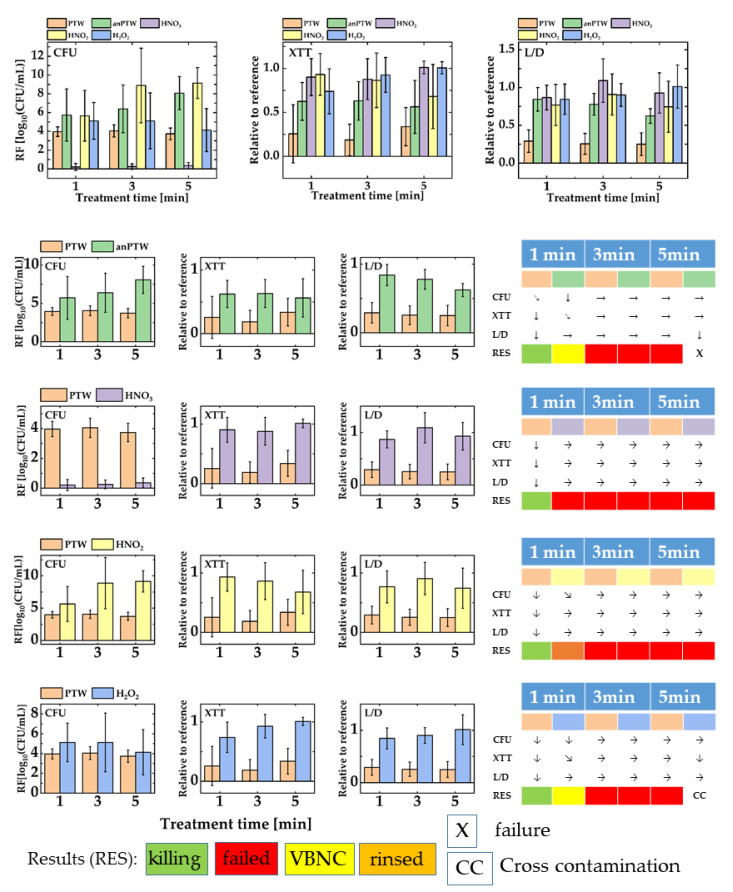
A summary of all assay data (CFU, XTT, and L/D) for biofilms of *Pseudomonas fluorescence.* The upper part of the figure gathers all data from all solutions (PTW, anPTW, HNO_3_, HNO_2_, and H_2_O_2_). The lower part of the sketch compares every mixed solution versus freshly produced PTW. The result of treatment with such solutions has been underpinned with an assay evaluation system as described earlier (Materials and Methods Section 2.6.2). For convenience, only statistically meaningful changes have been taken into account. In a series of independent ANOVA (*p* < 0.5), the results of a certain treatment time have been compared with those of the previous point of the same data set (1 min vs. reference, 3 min vs. 1 min, and so on).

**Table 1 microorganisms-11-00932-t001:** Summary of traceable compounds found (via IC and potentiometric measurements) in freshly generated PTW. For an accurate determination of the NO_2_-concentration, possible chemical reactions such as oxygenation of NO_2_ to NO_3_ were impeded by the addition of NaOH.

Title 1	Concentration [mg/L]	Concentration [mol/L]
HNO_3_	72.3	1.17
HNO_2_	1600.7	33
H_2_O_2_	717.3	21.1

## Data Availability

The data can be offered upon request of interested readers.

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
