# Peer review of "Plasma-Treated Water: A Comparison with Analog Mixtures of Traceable Ingredients"

_microorganisms, 2023, doi:10.3390/microorganisms11040932_

Round 1

Reviewer 1 Report

 Plasma treated water: a comparison with analogue mixtures of 2 traceable ingredients

This manuscript is creative to use the PTW and anPTW to investigate the antibacterial mechanisms of PTW. However, several sections need to be addressed for suitable publication, particular the lack of description for the procedures of biofilm making, antibacterial tests, and controls. Additionally, many articles have presented and discussed the antibacterial mechanism of plasma active water, plasma treated water and plasma. Thus, discussion based on these articles is expected. This manuscript must address these for publication.  

Following is my comments:

1.     Introduction: Many researches used PAW to inactivate pathogens on food items. Some of them also measured concentrations HNO3-, HNO2-, and H2O2 as well as the analyses of pH, oxidative-reduction potential (ORP), and electron paramagnetic resonance (EPR) to measure the free radicals. Recommend authors to describe those researches.

2.     Line 76-84: It should belong to Introduction. Author also need to explain why choose microwave to generate plasma in Introduction.

3.     Line 85-93: Suggest authors present a figure of PAW. In addition, the description is not clear and lack some key information, such as power, what type of air, the distance of plasma torch to water surface, what type of water, etc…

4.     Line 100-101: author did not explain why use these parameters to generate plasma, particularly these parameters are different from previously describe in Line 85-93.

5.     Section 2.3: Oxidative-reduction potential (ORP) has been shown to be more critical than pH. Why did not authors check the ORP values of PTW?

6.     Line 161: “without any shaking”, why? P. fluorescence grows much better in aerobic condition.

7.     Section 2.4.1 and 2.3.2? How did authors grow biofilm and conduct antibacterial test? There is no description for these two parts. In addition, all experiments have control, positive and negative. However, there is no description of control groups.

8.     Line 170: 1:1012? It is very unusually to use this high dilution for colony counting.

9.     Line 179: “the CFU/mL of the reference” Does it mean the untreated control?

10.  Section 3.1: All nitrogen compounds are not correctly subscript.

11.  Fig. 3 and its description in text: it is difficult to understand “relative to the reference” and the units on Y axle mean since no explanation is presented.

12.  Did authors use the water treated with the air but without plasma as the reference or just water without being treated with air as the reference? This is the foundation of this research but authors did not describe.

13.  Some research demonstrated ozone being generated by plasma and exits in the plasma treated water. However, authors did not check the generation of ozone   nor discuss ozone.

14.  Since the antibacterial efficacy of PTW was higher than anPTW and HNO3, HNO2, and H2O2, it indicated that PTW contained other substances other than the aforementioned ones or PTW possessed some physical characteristics which do not exist in anPTW. Many articles have described these characteristics but authors did not present the discussion based on previous studies and this one.

Reviewer 2 Report

The paper deals with a fundamentally very important aspect, namely the analysis and possible standardization of the actual active components of plasma treated water by analyzing the principal chemical substances, their ratio and their involvement in the antimicrobial effect of the PTW.

However, there are a number of shortcomings that preclude publication of the manuscript in the form presented: 1. content errors ·       The goal of the study and the added value of the investigation for the field of science is not clear. ·       Material and methods: o   It is not clear how many repetitions were performed in total. This applies to the analysis of the PTW, especially to the microbiological inactivation tests. o   The description of the microbiological tests, including the production of the biofilms, is absolutely incomprehensible to me. o   In connection with biofilms, their structure and stability are of exceptional importance for the validity of the investigations. o   It is particularly serious that the actual treatment of the bacteria (suspended cells, biofilms) is not really described. o   It is not clear whether and how a standardization (mainly on the part of the bacteria) was carried out. However, this is essential in connection with the assessment of the effect of biocides.   2. formal errors ·       In the list of affiliations there is an Institution named “3”. However, this number cannot be found in the line of authors. ·       In the part Material and methods, the authors use present perfect/presence instead of simple past. Therefore, it is really difficult to recognize what they have really done versus what is known from literature.

·       In my opinion, there is no clear separation between the individual sections of the manuscript, i.e. there is detailed description of the methods and discussion in the results section, description of the literature in the part Material and methods (line 76-84) etc. Overall, this fact makes it extremely difficult to understand and follow the contents of the paper.

·       There is also a lot of redundancy in the manuscript. ·       Figures 3 and 4 are extremely confusing and do not help to understand the results.

Round 2

Reviewer 1 Report

Line 225 P. fluorescence should be italic.

Reviewer 2 Report

The manuscrpt was improved substantially.